

# Associations of 24-hour movement guidelines adherence with fruit and vegetable intake in university students

Yao Zhang[1], Xingyi Yang[2], Zhen Yang[3], Xinli Chi[4] and Sitong Chen[5]

[1] Physical Education Department, Zhengzhou Shengda University, Zhengzhou, Henan, China
[2] School of Physical Education, Shanghai University of Sport, Shanghai, Shanghai, China
[3] Department of Movement Sciences, KU Leuven, Leuven, Belgium
[4] School of Psychology, Shenzhen University, Shenzhen, Guangdong, China
[5] Centre for Mental Health, Shenzhen University, Shenzhen, Guangdong, China

Corresponding author
Sitong Chen,
sitong.chen@live.vu.edu.au

## ABSTRACT

**Background:** Unhealthy eating habits, such as low vegetable and fruit intake, are associated with many health problems. 24-h movement behaviors have been reported to be positively associated with numerous health-related outcomes. Despite the importance of these two modifiable lifestyle behaviors in building healthy habits in university students, there is a paucity of relevant research in this population. Therefore, this study aims to examine the correlation between compliance with 24-h movement guideline (24-h MG) and intake of fruits and vegetables (IFV) in Chinese university students.

**Methods:** This study investigated the relationship between the compliance with 24-h MG and IFV in 1,793 Chinese university students using a convenience sampling method online. Physical activity (PA) and sedentary behavior (SB) were assessed by the International Physical Activity Questionnaire-Short Form, while sleep was measured using the Pittsburgh Sleep Quality Index. The Chinese version of the Health Promoting Lifestyle Profile II was used to measure IFV. Generalized linear models were applied to examine the correlation between compliance with the 24-h MG and eating habits.

**Results:** The proportion of participants who routinely consumed vegetables and fruits was 24.6% and 43.1%, respectively, while the proportion of meeting the three 24-h MG and a combination of any two guidelines was 27.8% and 40.1%, respectively. Meeting all three guidelines was associated with a greater IFV intake compared to not meeting either guideline. Meeting all three guidelines (OR = 2.42 [1.63, 3.58]) and the combination of moderate to vigorous PA (MVPA) and sleep (OR = 2.06 [1.37, 3.10]) were positively associated with the frequency of vegetable consumption ($p < 0.05$). As well, meeting all three guidelines (OR = 2.06 [1.37, 3.10]), the combination of MVPA and sleep (OR = 1.72 [1.04, 2.84]), and sleep only (OR = 1.88 [1.21, 2.92]) were positively associated with fruits consumption ($p < 0.05$).

**Conclusion:** Almost a third of the university students met the three 24-h MG, and compliance with all three guidelines was associated with a higher frequency of IFV. Furthermore, meeting the sleep guideline alone or in conjunction with the PA, and meeting the entire 24-h MG was associated with greater consumption of fruits.

## INTRODUCTION

Eating habits are a dynamic entity influenced by familial, cultural, social, and economic factors, varying across time and geographical regions (*Chen & Antonelli, 2020*; *Leng et al., 2017*). University students, experiencing greater independence in shaping their lifestyle behaviors, are particularly prone to nutritional imbalances. This demographic often opts for energy-dense, high-fat, high-carbohydrate fast foods as their primary dietary choice (*Arroyo Izaga et al., 2006*; *López-Azpiazu et al., 2003*), leading to a reduced daily intake of fruits and vegetables (IFV) and, in some cases, fewer than three meals per day (*Arroyo Izaga et al., 2006*; *Durán, Castillo & Vio del, 2009*). However, the importance of fruit and vegetable consumption in maintaining a diverse and nutritious diet cannot be overstated (*Harris et al., 2023*). These food groups are vital sources of essential micronutrients, fiber, and antioxidants, which are crucial for physical and mental health (*Kaplan et al., 2007*; *Lhakhang et al., 2014*; *Omena Messias et al., 2016*). Antioxidants, including vitamin C and carotenoids, play a significant role in protecting the body against oxidative stress, a primary factor in neurodegenerative diseases, chronic inflammatory conditions, atherosclerosis, certain cancers, and some forms of depression (*Byers & Perry, 1992*; *Irshad & Chaudhuri, 2002*; *Raison & Miller, 2011*).

A low IFV is a significant, modifiable risk factor contributing to the global increase in chronic diseases, obesity, and other chronic degenerative conditions (*Dixon et al., 2004*; *Kamphuis et al., 2007*; *Paisley, Sheeshka & Daly, 2001*). Previous studies have documented a rising prevalence of overweight and obesity in adults, while emerging evidence highlights the positive impact of healthy diets on physical and mental well-being (*Robberecht, De Bruyne & Hermans, 2016*; *Rooney, McKinley & Woodside, 2013*). For example, a review study indicated that higher levels of IFV in adults can play roles in promoting mental health (*e.g.*, reducing risks of psychological distress and depressive symptoms) (*Głąbska et al., 2020*), controlling unhealthy weight status (*e.g.*, adiposity and obesity) (*Hebden et al., 2017*; *Schwingshackl et al., 2015*), cardiovascular disease and some specific cancers prevention (*Wang et al., 2014*; *Sun et al., 2021*). Adequate IFV intake is a critical component of a nutritious diet (*World Health Organization, 2005*), with fruit and vegetable consumption serving as an indicator of overall diet quality and energy intake. This makes a high intake of these foods a key factor in the diet-health relationship (*Lampe, 1999*; *Trichopoulou et al., 2003*). It is estimated that increasing IFV consumption could save approximately 2.7 million lives annually (*World Health Organization, 2005*). Moreover, two meta-analyses and a prospective cohort study have shown that increased fruit and vegetable intake can reduce risks associated with cardiovascular diseases, cancer, and all-cause mortality (*Wang et al., 2014*; *Aune et al., 2017*).

Physical activity (PA), sedentary behavior (SB), and sleep are recognized as independent determinants of health outcomes in various populations. Engaging in sufficient PA, minimizing SB, and ensuring adequate sleep are each independently linked to improved

health in adults and older adults (*Bull et al., 2020*; *Chaput et al., 2020*; *Saunders et al., 2020*). A lifestyle characterized by regular PA enhances total daily energy expenditure, potentially aiding in weight loss. PA is widely recommended for its energy expenditure benefits and is deemed crucial in preventing chronic diseases. The escalating global incidence of obesity underscores its significance as a major public health issue (*James, 2008*).

Existing research indicates a positive associaiton between SB and poor diet quality, alongside a negative association with healthy food intake (*Pearson & Biddle, 2011*; *Deforche et al., 2015*). Interestingly, one study highlighted a relationship between certain PA components and increased fruit and vegetable consumption (*Opoku-Acheampong et al., 2018*). Additionally, evidence points to a gender-specific positive correlation between PA and fruit and vegetable intake, noted particularly in girls (*Chaput et al., 2020*). There is also an observed link between extended sleep duration in childhood and higher fruit and vegetable consumption (*Moreira et al., 2010*).

While the Canadian 24-h Movement Guidelines (MG) for adults advocate integrating PA, SB, and sleep as a cohesive research approach (*Ross et al., 2020*), research focusing on this integrated paradigm in adult populations remains limited. Numerous studies have examined the relationship between adherence to the Canadian 24-h MG and various health outcomes in children and youth, including obesity measures (*Chen et al., 2021*; *López-Gil et al., 2020*, *2023*), physical fitness (*López-Gil et al., 2020*; *Shi et al., 2020*; *Tanaka et al., 2020*), mental health or disorders (*Brown, Cairney & Kwan, 2021*; *Sampasa-Kanyinga et al., 2020*, *2021*), health-related quality of life (*Khan, Lee & Tremblay, 2021*; *Sampasa-Kanyinga et al., 2017*), and academic performance (*Lien et al., 2020*; *Watson, Dumuid & Olds, 2020*; *Bao et al., 2024*; *Tapia-Serrano et al., 2022*; *Chen et al., 2024*). Additionally, the association between compliance with the 24-h MG and dietary habits has been explored. Some researchers, such as *Thivel et al. (2019)* and *da Costa et al. (2021a, 2021b)*, have investigated the association between 24-h MG adherence and dietary pattern, indicating the positive roles of the 24-h MG in healthy eating promotion. Further, one recent study by *Tapia-Serrano et al. (2022)* focused on the association between 24-h MG and specific eating habits, such as IFV, indicating that meeting all the integrative guidelines was more likely to consume fruit and vegetables once a day in adolescents. However, such evidence is limited to younger subpopulations instead of adults.

Evidence has indicated that compliance with the 24-h MG is associated with improved dietary habits in adolescent population (*Thivel et al., 2019*; *da Costa et al., 2021a, 2021b*). Although some studies have examined the factors influencing IFV (*Alkazemi & Salmean, 2021*; *Deliens et al., 2018*; *Mello Rodrigues et al., 2019*; *Adams & Colner, 2008*), there have been no studies to examine the combination of PA, SB and sleep with IFV in university or college students (*Alkazemi & Salmean, 2021*; *Deliens et al., 2018*; *Mello Rodrigues et al., 2019*; *Adams & Colner, 2008*). Also, whether the association between 24-h MG and dietary habits extends to other subpopulation groups remains unclear. Notably, there is a dearth of research examining these associations in university students, which are at a crucial stage for establishing lifelong health behaviors. Understanding these dynamics could offer valuable insights for interventions aimed at promoting healthier eating habits in this

demographic. Therefore, the aim of this study seeks to explore the association between adherence to the 24-h MG and IFV in a sample of Chinese university students.

## METHODS

### Study design and participants

This cross-sectional study was conducted *via* an online questionnaire platform (https://www.wjx.cn/) from August 21 to 31, 2020. A convenience sampling approach was employed to recruit Chinese college students. The recruitment was announced on social media platforms, including WeChat and QQ. Prior to participating, students gave their consent. The questionnaire took approximately 15 min to complete, and participants were compensated with 10 RMB *via* online payment for their participation. In this cross-sectional study, the inclusion criteria for study participants were Chinese university students aged 18 years and above who were currently enrolled in a university. Participants were required to complete the survey questionnaire in full. Exclusion criteria included individuals who did not provide written informed consent, those who were not currently enrolled as college students, and those who did not complete the questionnaire. Additionally, participants with known medical conditions or disabilities that could affect their physical activity, sedentary behavior, or dietary habits were excluded from the study. Out of the 1,942 students from 30 provinces and regions (predominantly from Guangdong Province) who completed the survey, 1,846 (95.2%) provided complete and valid responses, forming the study's analytical sample. The Human Research Ethics Committee of Shenzhen University (Approval No: 2020005) approved the entire study, including recruitment, data collection, and all other research procedures.

### Measures

#### 24-h movement behaviors

PA and SB were self-reported using the Chinese version of the International Physical Activity Questionnaire-Short Form (IPAQ-SF) (*Macfarlane et al., 2007*). This questionnaire asks about time spent walking, moderate PA (MPA), vigorous PA (VPA), and SB in the preceding 7 days. The aggregate of MPA and VPA duration was termed as moderate to vigorous PA (MVPA). The Chinese IPAQ-SF has been validated for its psychometric properties in the Chinese population (*Macfarlane et al., 2007*; *Hu et al., 2015*; *Tao et al., 2017*). Sleep duration was assessed using an item from the Chinese version of the Pittsburgh Sleep Quality Index (PSQI), inquiring about the average nightly sleep over the past 30 days (*Liu & Tang, 1996*). The PSQI is well-validated in Chinese populations (*Wu et al., 2019*; *Zhou et al., 2020*). Participants adhering to the Canadian 24-H MG for Adults aged 18–64 years–7 to 9 h of sleep per night, 8 h or less of SB per day, and a minimum of 150 min of MVPA per week–were considered compliant with the guidelines (*Ross et al., 2020*).

#### Intake of fruit and vegetable

Fruit and vegetable intake data were gathered using two items from the Chinese version of the Health Promoting Lifestyle Profile II (HPLP II) (*Zhou et al., 2022*). The intake

frequency was self-reported on a four-point Likert scale from "never" to "routinely", with higher scores indicating better intake. The HPLP II has demonstrated acceptable reliability and validity in the Chinese population (Cronbach's $\alpha$ = 0.86, $\chi^2$ = 42.91, CFI = 0.992, TLI = 0.987, RMSEA = 0.046, SRMR = 0.015) (*Chi et al., 2021*).

*Sociodemographic information (control variables)*
Sociodemographic data included gender, age, subjective socioeconomic status (on a scale from 1 for "the poorest" to 10 for "the richest"), and body mass index (BMI, calculated as weight in kilograms divided by height in meters squared). Participants self-reported all sociodemographic information *via* the questionnaire.

## Statistical analyses

Descriptive statistics were initially performed to describe sample characteristics. All variables, except body mass index (BMI) (described using mean and standard deviation), were reported as percentages. Logistic regression models were developed to examine the correlation between adherence to 24-h MG and various outcomes, controlling for the other outcome and all sociodemographic variables. Two sets of multinomial logistic regression models were constructed: one set examined the association between the number of 24-h MG met and outcomes, and the other one assessed the specific number of 24-h MG and outcomes. These models were analyzed using generalized linear models (GLMs) in SPSS Version 26.0, with a statistical significance level set at $p < 0.05$ (two-sided). Model results were presented as odds ratios (OR) with 95% confidence intervals (CI).

## RESULTS

The final analysis included 1,793 participants, with their descriptive characteristics detailed in Table 1. The average age was 20.7 ± 1.6 years, and the majority (63.6%) were female. Regarding vegetable consumption, 36.9% of participants indicated they ate vegetables 'sometimes', and 35.3% 'often'. In contrast, a smaller proportion, 3.2%, reported 'never' consuming vegetables, while 24.6% did so 'routinely'. For fruit intake, the majority reported 'often' (41.5%) and 'routinely' (43.1%) consumption, with only 1.6% and 13.8% indicating 'never' and 'sometimes' respectively.

In terms of adherence to the 24-h MG, 48.6% of participants met the PA guideline. Compliance with the SB and sleep guidelines was higher, at 70.6% and 70.3%, respectively. Notably, only 27.8% of participants met all three components of the 24-h MG, with the majority (40.1%) meeting two of the guidelines. The proportion of participants adhering to specific guidelines varied, with as few as 3.2% meeting only the PA guideline, and up to 27.8% meeting all three guidelines.

Table 2 illustrates the association between adherence to the 24-h MG (number of guidelines met) and the frequency of fruit and vegetable intake. Initially, a significant association was found only among individuals complying with all three guidelines, showing an increased frequency of vegetable (OR = 2.20, 95% CI [1.50, 3.25], $p < 0.001$) and fruit intake (OR = 1.91, 95% CI [1.28, 2.85], $p = 0.002$), compared to those not adhering to any guidelines. After adjusting for covariates, this association remained significant, with an increased frequency of vegetable (OR = 2.41, 95% CI [1.63, 3.58],

**Table 1 Sample characteristics of Chinese university students ($N = 1{,}793$).**

|  |  | n | Percent (%) |
|---|---|---|---|
| Sex |  |  |  |
|  | Male | 653 | 36.4 |
|  | Female | 1,140 | 63.6 |
| Age (mean ± standard deviation) |  | 20.7 ± 1.6 |  |
| Body mass index (mean ± standard deviation) |  | 20.3 ± 2.9 |  |
| Perceived family affluence (mean ± standard deviation) |  | 5.7 ± 1.6 |  |
| Vegetable intake per day |  |  |  |
|  | Never | 57 | 3.2 |
|  | Sometimes | 662 | 36.9 |
|  | Frequently | 633 | 35.3 |
|  | Always | 441 | 24.6 |
| Fruit intake per day |  |  |  |
|  | Never | 28 | 1.6 |
|  | Sometimes | 248 | 13.8 |
|  | Frequently | 744 | 41.5 |
|  | Always | 773 | 43.1 |
| Meeting the PA guidelines |  |  |  |
|  | Not meeting | 921 | 51.4 |
|  | Meeting | 872 | 48.6 |
| Meeting the SB guidelines |  |  |  |
|  | Not meeting | 527 | 29.4 |
|  | Meeting | 1,266 | 70.6 |
| Meeting the sleep guidelines |  |  |  |
|  | Not meeting | 532 | 29.7 |
|  | Meeting | 1,261 | 70.3 |
| Number of guidelines met |  |  |  |
|  | Meeting none | 109 | 6.1 |
|  | Meeting any one | 467 | 26.0 |
|  | Meeting any two | 719 | 40.1 |
|  | Meeting all | 498 | 27.8 |
| Specific combinations of guidelines met |  |  |  |
|  | None | 109 | 6.1 |
|  | PA only | 58 | 3.2 |
|  | SB only | 166 | 9.3 |
|  | Sleep only | 243 | 13.6 |
|  | PA + SB | 199 | 11.1 |
|  | PA + Sleep | 117 | 6.5 |
|  | SB + Sleep | 403 | 22.5 |
|  | All | 498 | 27.8 |

**Note:**

PA, physical activity; SB, sedentary behavior.

**Table 2 Associations of number 24-h movement guidelines meet with vegetable and fruit intake in Chinese university students (N = 1,793).**

| Number of guidelines met | Vegetable intake | | | | Fruit intake | | | |
|---|---|---|---|---|---|---|---|---|
| | OR | 95% CI | | p value | OR | 95% CI | | p value |
| Unadjusted | | | | | | | | |
| 3 | **2.20** | **1.50** | **3.25** | **0.000** | **1.91** | **1.28** | **2.85** | **0.002** |
| 2 | 1.44 | 0.99 | 2.10 | 0.055 | 1.26 | 0.85 | 1.86 | 0.253 |
| 1 | 1.14 | 0.77 | 1.68 | 0.517 | 1.42 | 0.95 | 2.13 | 0.089 |
| 0 | Reference | | | | Reference | | | |
| Adjusted | | | | | | | | |
| 3 | **2.41** | **1.63** | **3.58** | **0.000** | **2.05** | **1.37** | **3.09** | **0.001** |
| 2 | 1.45 | 0.99 | 2.11 | 0.056 | 1.26 | 0.85 | 1.86 | 0.252 |
| 1 | 1.17 | 0.79 | 1.73 | 0.438 | 1.45 | 0.96 | 2.17 | 0.075 |
| 0 | Reference | | | | Reference | | | |

**Note:**
Adjusted model controlled for all the covariates this study included. Reference: reference group. OR, Odds Ratio; 95% CI, 95% Confidence Interval; Bold values indicate $p < 0.05$.

**Table 3 Associations of specific combinations of 24-h movement guidelines meet with vegetable and fruit intake in Chinese university students.**

| | Vegetable intake | | | | Fruit intake | | | |
|---|---|---|---|---|---|---|---|---|
| | OR | 95% CI | | p value | OR | 95% CI | | p value |
| Unadjusted | | | | | | | | |
| All | **2.21** | **1.50** | **3.25** | **0.000** | **1.91** | **1.28** | **2.86** | **0.002** |
| SB + Sleep | 1.39 | 0.94 | 2.06 | 0.101 | 1.25 | 0.83 | 1.87 | 0.294 |
| PA + Sleep | **1.64** | **1.02** | **2.66** | **0.043** | **1.68** | **1.02** | **2.76** | **0.043** |
| PA + SB | 1.45 | 0.94 | 2.24 | 0.096 | 1.08 | 0.69 | 1.69 | 0.739 |
| Sleep only | 1.25 | 0.82 | 1.90 | 0.306 | **1.85** | **1.19** | **2.87** | **0.006** |
| SB only | 0.94 | 0.59 | 1.47 | 0.772 | 0.95 | 0.60 | 1.52 | 0.840 |
| PA only | 1.33 | 0.74 | 2.37 | 0.343 | 1.48 | 0.80 | 2.72 | 0.212 |
| None | Reference | | | | Reference | | | |
| Adjusted | | | | | | | | |
| All | **2.42** | **1.63** | **3.58** | **0.000** | **2.06** | **1.37** | **3.10** | **0.001** |
| SB + Sleep | 1.37 | 0.92 | 2.03 | 0.118 | 1.23 | 0.82 | 1.85 | 0.324 |
| PA + Sleep | **1.71** | **1.05** | **2.77** | **0.030** | **1.72** | **1.04** | **2.84** | **0.034** |
| PA + SB | 1.46 | 0.94 | 2.27 | 0.090 | 1.10 | 0.70 | 1.73 | 0.685 |
| Sleep only | 1.28 | 0.84 | 1.95 | 0.260 | **1.88** | **1.21** | **2.92** | **0.005** |
| SB only | 0.96 | 0.61 | 1.52 | 0.860 | 0.97 | 0.61 | 1.55 | 0.896 |
| PA only | 1.38 | 0.77 | 2.48 | 0.287 | 1.53 | 0.83 | 2.84 | 0.177 |
| None | Reference | | | | Reference | | | |

**Note:**
PA, physical activity; SB, sedentary behavior; Reference: reference group. Adjusted model controlled for all the covariates this study included. OR, Odds Ratio; 95% CI, 95% Confidence Interval; Bold values indicate $p < 0.05$.

$p < 0.001$) and fruit intake (OR = 2.05, 95% CI [1.37, 3.09], $p = 0.001$) in individuals adhering to all three guidelines.

Further analyses of the association between specific combinations of adherence to 24-h MG and IFV are presented in Table 3. Compared to individuals not following any of the guidelines, those adhering to all three showed a significantly higher frequency of vegetable intake (OR = 2.42, 95% CI [1.63, 3.58], $p < 0.001$). Notably, adherence to both the MVPA and sleep guidelines was also associated with a higher frequency of vegetable intake (OR = 1.71, 95% CI [1.05, 2.77], $p = 0.030$). Similarly, a higher frequency of fruit intake was significantly associated with compliance to all three guidelines (OR = 2.06, 95% CI [1.37, 3.10], $p = 0.001$), adherence to both MVPA and sleep guidelines (OR = 1.72, 95% CI [1.04, 2.84], $p = 0.034$), and adherence to the sleep guideline alone (OR = 1.88, 95% CI [1.21, 2.92], $p = 0.005$).

## DISCUSSION

The primary objective of this study was to explore the association between adherence to 24-h MG and vegetables and fruits intake among Chinese university students. Our results demonstrate a positive association between full compliance with the 24-h MG and increased consumption of both vegetables and fruits. Specifically, adherence to all the 24-h MG was linked to higher odds of more vegetable and fruit intake. Additionally, adherence to both PA and sleep guidelines was associated with improved intake of these food groups, and adherence to sleep guidelines in insolation was associated with increased fruit intake. These findings are further discussed below.

The study revealed that only 27.8% of the participants met all three components of the 24-h MG, with a notably low adherence of 3.2% to the PA guideline alone. Previous research has reported varying prevalence rates of adherence to all three 24-h MG among adults in different regions: 25% in Slovenia (*Kastelic et al., 2021*), 18% in Chile (*Riquelme et al., 2022*), and 1.6% in Latin America (*Ferrari et al., 2022*). The lower prevalence rate observed in our study among Chinese university students could be even more pronounced if PA adherence were assessed on a daily basis. It is important to note that differences in prevalence rates across studies may be attributed to variations in assessment methods, such as the use of accelerometers *vs.* self-reported questionnaires for PA (*Rollo, Antsygina & Tremblay, 2020*; *Roman-Viñas et al., 2016*; *Tapia-Serrano et al., 2022*). Caution is advised in making direct comparisons across different studies due to these methodological differences. Nevertheless, the insights gained from this study are crucial for designing effective interventions aimed at enhancing adherence to the 24-h MG among Chinese adults, considering the substantial health benefits associated with these guidelines.

The findings of this study resonate with the suggestions of the Canadian 24-h MG for Adults, which posits that adherence to a full MG can yield additional benefits (*Ross et al., 2020*; *López-Gil et al., 2020*). Our study underscores a significant link between adherence to the 24-h MG and increased consumption of vegetables and fruits, supporting exist literature (*Thivel et al., 2019*), leaving a gap in understanding the specific relationship between the 24-h MG and particular dietary habits like vegetable and fruit consumption.

In the literature, to our knowledge, there is only one study that examined the association between 24-h MG and IFV, especially in adolescents. Findings of that study can in part support our results given the consistent associations between 24-h MG and fruit/vegetables consumption observed in each study regardless of the studied populations. This further highlights the importance of 24-h movement behaviors promotion in healthy eating habits.

There is sufficient evidence that adhering to a greater number of movement behaviors (*e.g.*, following three guidelines *vs.* two or one is associated with enhanced health benefits (*Cliff et al., 2017*; *Guimarães et al., 2021*). The VIRTUE framework emphasizes the importance of studying he connections between time-use behaviors and a wide array of health outcomes (*Pedisic, Dumuid & Olds, 2017*). Several studies have linked high levels of MVPA (*Södergren et al., 2008*), low SB (*Wilson et al., 2019*) and adequate sleep duration (*Shankar, Charumathi & Kalidindi, 2011*) with better self-rated health. Additionally, a positive relationship between compliance with 24-h GM and self-rated level of health and IFV in adults has been documented (*Rollo, Antsygina & Tremblay, 2020*). Furthermore, various combination of PA and diet patterns have been found to be predictive of longevity, suggesting a synergistic effect between diet and activity (*Nicklett et al., 2012*). These insights are crucial for public health promotion, highlighting the need to encourage adults to adhere to as many movement behavior guidelines as possible. Promoting an integrated approach to 24-h MG, as suggested by *Ross et al. (2020)*, *Jurakić & Pedišić (2019)*, could be an effective strategy. Considering that our study is one of the few that delve into the association between the 24-h MG and specific dietary habits, further research is needed. Future studies should aim to either corroborate or challenge our findings.

One important finding in our study is the association between adherence to recommended sleep guidelines individually and higher odds of more fruit consumption among Chinese university students. Our research reinforce the concept that adequate sleep, along with a balanced diet, is an essential component of a healthy lifestyle. Prior research has established a positive link between IFV and sleep duration. For instance, *Grandner et al. (2013)* reported an association between normal sleep duration and a more diverse diet. Studies have consistently found that individuals with shorter sleep duration tend to consume fewer fruits and vegetables (*Buxton et al., 2009*; *Kruger et al., 2014*; *Tatone-Tokuda et al., 2012*; *Westerlund, Ray & Roos, 2009*), whereas those with adequate sleep are more likely to have higher IFV (*Kruger et al., 2014*; *Hoefelmann et al., 2012*). Furthermore, aspects of inadequate sleep have been correlated with lower IFV, even after controlling for other predictors of sleep quality (*Jansen et al., 2021*).

However, much of the existing literature focuses on the relationship between sleep patterns and fruit/vegetable consumption in specific demographic groups such as employees, adolescents, and pregnant women (*Kruger et al., 2014*; *Hoefelmann et al., 2012*; *Duke et al., 2017*). Nonetheless, there is an urgent need for more research focusing on China's large population of university students. Other research has linked adherence to recommended sleep guidelines with reduced stress levels (*Kastelic et al., 2021*). Factors such as short sleep duration have been associated with increased impulsivity and mood

disturbances (*Rossa et al., 2014*), which can influence dietary choices and the motivation to maintain a healthy diet (*Fong et al., 2019*; *Izydorczyk et al., 2019*). Given these findings, it is clear that maintaining a sufficient duration of sleep is a key factor in promoting healthy dietary behaviors in young adults. Moreover, these insights suggest that dietary interventions could be more effective when combined with sleep hygiene strategies (*Irish et al., 2015*). This integrative approach could potentially enhance overall health and well-being in university students, a demographic that is particularly prone to irregular sleep patterns and dietary habits.

The current study offers valuable insights into the association between 24-h movement behaviors and IFV among university students, a demographic that has not been extensively studied in this context. The use of a large sample size of young adults strengthens the generalization of our findings. However, the study has several limitations that warrant to be addressed in future research. Firstly, the cross-sectional nature of the study limits our ability to draw conclusions about the temporal or causal relationship. Furthermore, the study's focus on university students, while providing valuable data on this specific group, does not necessarily reflect the behaviors of all generally healthy young adults in real-world settings. Another significant limitation is the reliance on self-reported questionnaires to measure 24-h movement behaviors and IFV. While the questionnaires used are widely accepted and have been extensively utilized in previous research, they are susceptible to recall biases. Participants' recall on their activities and dietary intake may not always be accurate, potentially introducing measurement errors. Additionally, while our study focused on 24-h movement behaviors, it did not delve into other potentially influential factors, such as sleep quality. These limitations may influence the evidence quality of associations observed in this study. Thus, higher quality evidence regarding the associations between 24-h MG and IFV can be gained through if addressing these limitations, including using longitudinal design, device-based measures, and considering more variables related to lifestyle behaviors.

## CONCLUSION

This study revealed that approximately one-third of university students in China successfully adhered to the 24-h MG for health promotion. A key finding is the positive association between adherence to all three components of the 24-h MG and a higher frequencies of IFV. Notably, adherence to the sleep guideline, whether in isolation or in combination with other components of the 24-h MG, is specifically linked to increased fruit intake.

These findings highlight the significant role of comprehensive adherence to movement behavior guidelines in promoting healthy dietary habits among young adults. The study particularly emphasizes the critical importance of adequate sleep in fostering better IFV. In light of these results, health promotion strategies targeting university students should focus on encouraging adherence to a full spectrum of movement behaviors, with a special emphasis on the role of sleep. By doing so, these strategies may effectively enhance the overall health and well-being of this population.

### Funding

The authors received no funding for this work.

### Competing Interests

The authors declare that they have no competing interests.

### Author Contributions

- Yao Zhang conceived and designed the experiments, performed the experiments, analyzed the data, prepared figures and/or tables, authored or reviewed drafts of the article, and approved the final draft.
- Xingyi Yang performed the experiments, prepared figures and/or tables, and approved the final draft.
- Zhen Yang performed the experiments, authored or reviewed drafts of the article, and approved the final draft.
- Xinli Chi analyzed the data, authored or reviewed drafts of the article, and approved the final draft.
- Sitong Chen conceived and designed the experiments, analyzed the data, prepared figures and/or tables, and approved the final draft.

### Human Ethics

The following information was supplied relating to ethical approvals (*i.e.*, approving body and any reference numbers):

Shenzhen University.

### Data Availability

The raw measurements are available in the Supplementary Files.

### Supplemental Information

Supplemental information for this article can be found online at http://dx.doi.org/10.7717/peerj.17875#supplemental-information.

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
