# Peer review of "Associations of 24-hour movement guidelines adherence with fruit and vegetable intake in university students"

_PeerJ, doi:10.7717/peerj.17875_

## Round 0.1 · original submission · Major Revisions

Please consider carefully the comments of the Reviewers.

Reviewer 1 ·

Basic reporting

no comment

Experimental design

no comment

Validity of the findings

no comment

Additional comments

entitled Associations of 24-hour movement guidelines adherence with fruits and vegetables intake in university students. I would also like to thank the authors for their effort and dedication in the preparation of the article. This paper presents an investigation of adherence to the three behaviours of the 24-Hour Movement Guidelines and their relationship to fruit and vegetable intake in university students. Overall, the article is well written and represents a novel and original advance in the study of the 24-Hour Guidelines. However, I would like to make some contributions to the authors that I think will help them to improve the quality of their work:
Lines 51: add the reference at the end of the sentence.
Line 56: add reference at the end of the sentence.
References 24, 25, 27 and 28 are for children and adolescents, I suggest you update them to university literature.
Line 94: “including obesity measures [32-34],” this reference is incorrect here, as it is not about 24-Hour Movement Guidelines. I suggest you replace reference 32 with:

1. López‐Gil, J. F., Tapia‐Serrano, M. A., Sevil‐Serrano, J., Sánchez‐Miguel, P. A., & García‐Hermoso, A. (2023). Are 24‐hour movement recommendations associated with obesity‐related indicators in the young population? A meta‐analysis. Obesity, 31(11), 2727-2739. https://doi.org/10.1002/oby.23848
Línea 95, en los indicadores de rendimiento académico, incluya las referencias:
1. Bao, R., Qin, H., Memon, A. R., Chen, S., López-Gil, J. F., Liu, S., ... & Cai, Y. (2024). Is adherence to the 24-h movement guidelines associated with greater academic-related outcomes in children and adolescents? A systematic review and meta-analysis. European Journal of Pediatrics, 1-12.
2. Tapia-Serrano, Miguel Angel, et al. "Is adherence to 24-Hour Movement Guidelines associated with a higher academic achievement among adolescent males and females?." Journal of Science and Medicine in Sport 25.2 (2022): 155-161. https://doi.org/10.1016/j.jsams.2021.09.005
Lines 99 - 101: synthesise the information from the Thivel et al. (2019) and Costa et al. (2021) studies in one sentence. In addition, I suggest you include the following study, as it is the most current study of the 24-Hour Movement Guidelines and dietary patterns:
1. Tapia-Serrano, M. A., Sánchez-Miguel, P. A., Sevil-Serrano, J., García-Hermoso, A., & López-Gil, J. F. (2022). Is adherence to the 24-Hour Movement Guidelines associated with Mediterranean dietary patterns in adolescents?. Appetite, 179, 106292. https://doi.org/10.1016/j.appet.2022.106292
Line 104: Add references after: “adoescent population”.
The authors say that there is no study that has looked at the relationship between 24-hour GM and fruit and vegetable intake. However, this is not entirely true. If they consult Table 2 of the previous study (Tapia-Serrano et al. 2022), the authors can verify that adherence to 24-hou GM and fruit and vegetable intake has been demonstrated in adolescents. Therefore, I think that the relevance of your study lies in the fact that it is the first study to analyse this relationship in university students, as there is a previous study that has shown this association in adolescents. Therefore, I suggest you take this into account and introduce the contribution of your study in comparison to the study by Tapia-Serrano (2022) prior to the objectives of your study. This is also interesting to discuss the findings of your study in the discussion section as I mark them below.

Line 148: replace control variables with covariates.
Line 162: put p in italics.
In table 1 include the mean values and standard deviations of the three behaviours of the 24-Hour Movement Guidelines. Also add at the end of the table notes of the explanation of PA.
Lines 184, 186 and 187: put the p-value in italics.
The title of Table 2, replace 24-hour movement guidelines with 24-hour MG. Also put the p-value in italics (apply this change to the whole document).
The title of Table 3, replace 24-hour movement guidelines with 24-hour MG.
Lines 227-228: this sentence: “Nevertheless, the association between the 24-hour MG and specific dietary habits, such as the consumption of vegetables and fruits, remains unstudied” is not correct. As I indicated above, there is a previous study that has shown this association in adolescents (Tapia-Serrano et al., 2022). I suggest you compare your results with Tapia-Serrano's findings.
Line 231: add a space before [63, 64].
Line 256: use the abbreviation VIRTUE directly.
Line 259: use the abbreviations MVPA and SB directly.
Line 261: topographical error: activity maybe they mean physical activity?
Line 276: use the abbreviation IFV directly.
Line 276: perhaps they meant Grandner et al [72].
Line 298: use the abbreviation IFV directly.
I suggest you thoroughly review the discussion as it seems that paragraph 3 and paragraph 5 of this section are duplicated.
Regarding the references, I suggest you add the doi in as many as possible. In addition, I have detected errors in some of them:
Dumuid D, Olds T. INTEGRATING SLEEP, SEDENTARY BEHAVIOUR, AND PHYSICAL ACTIVITY RESEARCH IN THE EMERGING FIELD OF TIME-USE EPIDEMIOLOGY: DEFINITIONS, CONCEPTS, STATISTICAL METHODS, THEORETICAL FRAMEWORK, AND FUTURE DIRECTIONS. Kinesiology 2017, 556 49:252-269.
I also suggest that you put the titles of studies in lower case.

Annotated reviews are not available for download in order to protect the identity of reviewers who chose to remain anonymous.

·

Basic reporting

The reviewed manuscript aimed to evaluate the associations of 24-hour movement guidelines adherence with fruits and vegetables intake in university students.
Generally speaking, the novelty is less enough, however, the authors have given an entire study about the examine the correlation between compliance with 24-hour movement guideline (24-hour MG) and intake of fruits and vegetables (IFV) in Chinese University students.

The manuscript is written in clear language, with sufficient references included. It provides a clear, concise, and direct background on the subject matter.

I have thoroughly reviewed all raw data files, ethical letters and found them to be satisfactory, with no flaws in data management.

Some points should be corrected and clarified. The details as below,

1.The background is well-developed and highlights the importance of healthy eating habits, PA, SB, and sleep in maintaining overall health. However, it could be strengthened by providing more specific examples or statistics related to the impact of unhealthy eating habits and the benefits of healthy behaviors.

2.The literature review is thorough and provides a good overview of previous research on the 24-hour MG, PA, SB, and sleep in relation to health outcomes. However, it could be more focused on studies relevant to university students and the specific relationship with IFV.

3. The authors should double check the citation format in main body of manuscript.

Experimental design

In a cross-sectional study, it is essential to clearly define both the inclusion and exclusion criteria to ensure the study sample is representative and the results are valid. The inclusion criteria should specify the characteristics participants must possess to be eligible for the study, while the exclusion criteria should specify the characteristics that disqualify individuals from participating. By clearly defining these criteria, researchers can ensure that the study sample accurately reflects the population of interest and that the results are reliable and generalizable. What was yours should be mentioned in the study?

Validity of the findings

Although the study is not new in terms of its purpose.
However, I believe, the study findings summarize and provides valuable insights for future research and health promotion strategies targeting university students in China.
The study focused on university students in China, which may limit the generalizability of the findings to other populations. Future studies should aim to replicate the findings in diverse populations to enhance the generalizability of the results.

In the Discussion section, the content should focus on comparing the obtained results with previous findings. Authors should emphasize relevant factors and avoid discussing irrelevant content. E.g., Line 284, 285, same for others.

The discussion acknowledges the limitations of the study, such as the cross-sectional design and reliance on self-reported questionnaires. However, further discussion on the potential impact of these limitations on the study's findings and their implications would enhance the discussion.

Additional comments

Some other minor revisions should be addressed,

Consider including a flow diagram to depict the student’s selection process, including screening, enrollment, and follow-up, to enhance transparency and clarity. It’s just a suggestion.

Heading number should not be insert in all main section from introduction to conclusion (e.g 1, 2…).

Don’t use subheading numbers in throughout the manuscript.
e.g., 2.1 Study design and participants, 2.2.1 24-hour movement behaviors
same for others.

Line 83 The word "associaiton" should be "association."
Line 228 This limitation may hinder the comparability of our findings with prior research." - The word "comparability" should be "comparison."
Line 103 This body of literature suggests that compliance with the 24-hour MG…The sentence should be rewrite.
Line 190 Table 3 delves into specific combinations of adherence to the 24-hour MG….the sentence is poor should be improved.
191 variables, post-covariate adjustment.
Line 229 "Scientific evidence has proposed that adhering to the recommendations of following more movement behaviors (e.g., 3 versus 2 or 2 versus 1) was linked with improved health benefits." - This sentence should be revised for clarity, perhaps as: "Scientific evidence suggests that adhering to more recommendations for movement behaviors (e.g., 3 versus 2 or 2 versus 1) is linked to improved health benefits."
Line 224 The word "associaiton" should be "association."
Line
Line 235 What do you mean by self-rated health?
Line 235 Why does it mention the self-rated health in this section? Is it part of the
objective of the research?
Line 251 You mentioned that, “This gap poses a challenge in directly comparing our findings with previous research” it should be better to provide a valid reference here?
Line 271-272 The sentence is a bit unclear and could be improved for better clarity. It seems like you're trying to convey that adherence to recommended sleep guidelines is associated with a lower frequency of reduced fruit consumption.
Line 286 Yet, studies specifically exploring this relationship among Chinese university students…. Sentence should be revised.
Table captions are not satisfactory right now; they should be improved if possible. To improve the table captions, you can make them more descriptive and specific.
For table title, instead of just stating the content of the table, you can provide more context or highlight key findings. Here are some revised table titles:
Original: Table 1 - Participant Characteristics
Revised: Table 1 – Demographic and sample Characteristics of Chinese University Students (N=1,793).
Same for the others….
In Table 1 authors mentioned n, % and below they elaborate it as
Age (mean ± standard deviation) 20.7 1.6
Body mass index (mean ± standard deviation) 20.3 2.9
Perceived family affluence (mean ± standard deviation) 5.7 1.6
I think the use of "%" and "SD" may cause confusion here for readers. Can you please clarify this?

---

## Round 0.2 · accepted · Accept

The authors have properly revised the manuscript as suggested by the Reviewers.

·

Basic reporting

I agree with the positive feedback provided in the review report comments. The authors have presented a well-executed study that offers new insights into the associations between adherence to 24-Hour Movement Guidelines and fruit and vegetable intake among university students. This research has the potential to inform future interventions aimed at improving dietary behaviours through adherence to movement guidelines. I recommend this manuscript for further process with no revisions needed.
Again, Thank you for the opportunity to review this valuable contribution to the field.

Experimental design

My comments and suggestions have been properly addressed. I have no further questions.

Validity of the findings

My comments have been properly addressed. I have no further questions.